# Hexylammonium Acetate-Regulated Buried Interface for Efficient and Stable Perovskite Solar Cells

**DOI:** 10.3390/nano14080653

**Published:** 2024-04-09

**Authors:** Ruiyuan Hu, Taomiao Wang, Fei Wang, Yongjun Li, Yonggui Sun, Xiao Liang, Xianfang Zhou, Guo Yang, Qiannan Li, Fan Zhang, Quanyao Zhu, Xing’ao Li, Hanlin Hu

**Affiliations:** 1Jiangsu Provincial Engineering Research Center of Low-Dimensional Physics and New Energy & School of Science, Key Laboratory for Organic Electronics and Information Displays & Institute of Advanced Materials (IAM), Jiangsu National Synergistic Innovation Center for Advanced Materials (SICAM), Nanjing University of Posts and Telecommunications, Nanjing 210023, China; 1221066812@njupt.edu.cn (T.W.); 1222067110@njupt.edu.cn (Y.L.); 1221066811@njupt.edu.cn (Y.S.); 1023082010@njupt.edu.cn (F.Z.); 2Hoffmann Institute of Advanced Materials, Shenzhen Polytechnic University, 7098 Liuxian Boulevard, Shenzhen 518055, China; 321087@whut.edu.cn (F.W.); 307432@whut.edu.cn (X.L.); 257508@whut.edu.cn (X.Z.); 22280078@mail.szpu.edu.cn (G.Y.); liqiannan@szpu.edu.cn (Q.L.); 3State Key Laboratory of Advanced Technology for Materials Synthesis and Processing, School of Materials Science and Engineering, Wuhan University of Technology, Wuhan 430070, China; cglamri@whut.edu.cn

**Keywords:** ionic liquids, perovskite devices, photovoltaic performance, interface defect states

## Abstract

Due to current issues of energy-level mismatch and low transport efficiency in commonly used electron transport layers (ETLs), such as TiO_2_ and SnO_2_, finding a more effective method to passivate the ETL and perovskite interface has become an urgent matter. In this work, we integrated a new material, the ionic liquid (IL) hexylammonium acetate (HAAc), into the SnO_2_/perovskite interface to improve performance via the improvement of perovskite quality formed by the two-step method. The IL anions fill oxygen vacancy defects in SnO_2_, while the IL cations interact chemically with Pb^2+^ within the perovskite structure, reducing defects and optimizing the morphology of the perovskite film such that the energy levels of the ETL and perovskite become better matched. Consequently, the decrease in non-radiative recombination promotes enhanced electron transport efficiency. Utilizing HAAc, we successfully regulated the morphology and defect states of the perovskite layer, resulting in devices surpassing 24% efficiency. This research breakthrough not only introduces a novel material but also propels the utilization of ILs in enhancing the performance of perovskite photovoltaic systems using two-step synthesis.

## 1. Introduction

The progress in achieving the power conversion efficiency (PCE) in perovskite solar cells (PSCs) has been strikingly remarkable, soaring from 3.8% in 2009 to an impressive 26.1% [1]. Achieving an even higher PCE has been a major focus of research, and one key avenue for improvement involves optimizing the electron transport layer (ETL) in n-i-p PSCs [2,3,4]. Among the commonly investigated ETLs, TiO_2_ has played a pioneering and thoroughly researched role in the development of PSCs [5,6]. However, TiO_2_ is not without its limitations.

Frequently, it faces challenges due to substantial photocatalytic activity and limited electron mobility, both of which can negatively impact the lifespan of PSCs [7,8]. Additionally, the complex preparation process for TiO_2_ can be a drawback [9,10]. Consequently, tin dioxide (SnO_2_) has surfaced as a promising alternative ETL material owing to its diverse range of advantages, including the ability to be processed at lower temperatures and its capacity for high power-conversion efficiency [11]. Establishing an understanding of the correlation characteristics of SnO_2_ and the performance exhibited by PSCs is essential to fully exploit its potential as an ETL [12,13,14]. Despite the aforementioned notable advantages, attention should be paid to the presence of oxygen vacancies, dangling hydroxyl groups (-OH) and under-coordinated metal atoms in the SnO_2_ film. These factors may result in the accumulation of charge carriers and losses due to non-radiative recombination [15]. Furthermore, the concealed interface between the perovskite and SnO_2_ plays a pivotal role in attaining high-performance PSCs. The existence of halogens, organic ions and under-coordinated metals at this buried interface can instigate interfacial chemical reactions [15,16,17]. The concealed interface also holds significant sway over the crystal growth dynamics of the upper perovskite films [18,19,20]. Hence, modifying the interface between SnO_2_ and perovskite is crucially important to improve both the PCE and stability of the device [21,22,23]. Addressing the challenges associated with SnO_2_ while leveraging its unique advantages offers significant potential for advancing the realm of perovskite solar cells.

In recent years, significant efforts have been devoted to focusing on countering the development of interfacial defects by employing functional materials for interface passivation. This research encompasses two key dimensions. The first dimension explores the use of inorganic compounds, primarily exemplified by alkali metal compounds, with potassium ions (K^+^) emerging as effective additives. K^+^ has showcased its potential in addressing issues such as mitigating boundary defect states, the hysteresis of *J*–*V* curves, and reducing ion migration. Collectively, these improvements lead to enhanced PCE in SnO_2_-based planar PSCs [24,25,26]. The second dimension delves into the role of organic molecules equipped with specific functional groups like −COO^−^, −SO_3_^−^, and −NH^3+^ [27,28,29]. These organic compounds wield significant influence over the surface energy levels of SnO_2_, ideally resulting in the suppression of surface trap states [30].

The properties of ionic liquids (ILs) position them as environmentally friendly materials known for their low toxicity, non-volatile properties, and exceptional solvent capacity [31]. These attributes render them a favored choice for their application in PSCs [32,33]. Kang and Zhang’s group innovatively devised an IL-based buried interface modification strategy. This approach resulted in enhanced control over perovskite crystal growth, significantly improving crystal quality, notably decreasing non-radiative recombination losses attributed to interface defects and enhancing energy level alignment. These achievements are made possible by effectively passivating interfacial defect states and adjusting the energy band structure. This results in a decrease in carrier transport barriers. Implementing this strategy, they engineered all-inorganic CsPbI_3_ photovoltaic devices, achieving an unprecedentedly low open-circuit voltage (*V*oc), a loss of 0.451 V, and an impressive PCE of 20.98% [34]. Zang’s team, on the other hand, employed a straightforward and highly efficient universal strategy for anion modification, which was implemented to strengthen the photovoltaic performance of devices. This approach encompasses the introduction of a range of guanidinium salts containing diverse anions to regulate the perovskite/SnO_2_ interface. Each of these anions serves an essential function in defect passivation on both SnO_2_ and the perovskite layer, optimizing interface band alignment and enabling the crystallization of PbI_2_ and perovskite. Ultimately, this led to the achievement of the highest PCE, reaching 23.74%, significantly surpassing the control device at 21.84%. Notably, this interface modification also enhanced both thermal stability and environmental resilience [35]. Owing to the advantages demonstrated by ILs in optimizing the hole transport layer (HTL)/perovskite interface, the search for highly efficient and practical ILs has now emerged as a foremost priority in this field. For the current mainstream preparation methods of perovskite, the one-step method has proven to be efficient in producing smooth and compact perovskite films. However, the antisolvent process, while effective, is difficult to control and unsuitable for fabricating large-area devices, often resulting in poor surface coverage and diminished cell performance [28,36]. In contrast, the utilization of a two-step deposition method offers a more effective and manageable solution. This method involves the separate deposition of the organic layer and PbI_2_ layer.

In this study, we introduced hexylammonium acetate (HAAc) integrated into the interface between SnO_2_ and perovskite to enhance device performance. The introduction of HAAc initiates chemical bonding between Ac^-^ and SnO_2_, effectively filling oxygen vacancies through the creation of hydrogen bonds. This critical step results in a reduction in hydroxyl defects on the surface of SnO_2_, which would otherwise create potential barriers hindering carrier transport. Simultaneously, HA^+^ ions coordinate with uncoordinated Pb^2+^ ions within the perovskite material, effectively filling vacancies and, consequently, suppressing non-radiative recombination while boosting carrier mobility. Through the utilization of various testing techniques, we demonstrated that the reduction in defects leads to a notable reduction in non-radiative recombination, ultimately resulting in the enhancement of photovoltaic performance. The HAAc-treated device achieves an impressive peak PCE of 24.16% as a consequence, with a short-circuit current (*J*sc) of 25.49 mA cm^−2^, a *V*oc of 1.190 V, and a fill factor (FF) of 79.63%. In contrast, the control device with bare SnO_2_ possesses a lower PCE of 20.90%, accompanied by a *J*sc of 25.23 mA cm^−2^, *V*oc of 1.103 V, and FF of 75.27%. Furthermore, our study demonstrates the achievement of long-term stability. The HAAc-treated devices exhibit minimal degradation, maintaining an impressive 99.6% of original PCE after 1200 h.

## 2. Results and Discussion

This study involves the fabrication of planar n-i-p PSCs with the following structure: ITO/SnO_2_/HAAc/perovskite/Spiro-OMeTAD/Au, which is shown in detail in Figure 1a. Briefly, HAAc is deposited on the surface of SnO_2_ via spin-coating. Then, the perovskite layer is fabricated using a two-step method [37], and the schematic diagram for the preparation process is shown in Appendix A. Finally, the device is formed after depositing the HTL (spiro-OMeTAD) and Au electrode. The structural depiction of HAAc can be observed in Appendix A. The incorporation of HAAc serves as a pivotal strategy to tailor the SnO_2_/perovskite interface. From the cross-sectional scanning electron microscopy (SEM) images of the control device and HAAc-treated devices displayed in Figure 1b,c, it is evident that the HAAc-modified perovskite film displays a markedly more homogeneous morphology in comparison to the control films. This modified film boasts significantly larger perovskite grains and a reduced presence of grain boundaries compared to the unmodified film. In Figure 1d, we delve into the underlying mechanism of HAAc. HAAc comprises a cationic HA^+^ and an anionic Ac^-^. Upon the introduction of HAAc, Ac^-^ forges a chemical bond with SnO_2_, ushering in the occupancy of oxygen vacancies via the establishment of hydrogen bonds. As a result, this process mitigates the presence of hydroxyl defects on the SnO_2_ surface, thereby dismantling potential obstacles that obstruct the movement of charge carriers [38]. Simultaneously, HA^+^ adeptly coordinates with uncoordinated Pb^2+^ ions residing within the perovskite material, effectively occupying the vacancies and subsequently suppressing non-radiative recombination while bolstering the charge carrier mobility. Evidence from the Energy Dispersive Spectrometer (EDS) results unmistakably indicate the presence of HAAc on the surface of SnO_2_, with the nitrogen (N) component deriving from the amino group of the ionic liquid HAAc, as depicted in Appendix A.

The chemical interplay between HAAc and SnO_2_ is vividly illustrated through the utilization of X-ray photoelectron spectroscopy (XPS), the whole surveys of which are displayed in Appendix A. In Figure 2a, a conspicuous shift is observed in both the Sn 3d_5/2_ and Sn 3d_3/2_ peaks within the SnO_2_, transitioning from 495.1 eV and 486.7 eV to 494.8 eV and 486.4 eV, respectively. This shift in the Sn 3d signal in the direction of lower binding energies implies a reduction in the cationic charge of uncoordinated Sn^2+^ ions within the SnO_2_ structure. This phenomenon is likely due to the transfer of lone-pair electrons from the hydroxyl groups (-OH^-^) found in HAAc, resulting from a coordinated interaction. A comparable phenomenon is noted in the O 1 peaks, as represented in Figure 2b. In both the control and HAAc-treated samples, a broad and asymmetrical peak emerges, which encompasses lattice oxygen (OL) and vacancy oxygen (OV) within the SnO_2_, respectively. The extent of the vacancy oxygen (OV) peak area is quantified using the formula OV/(OV + OL), as demonstrated in Figure 2c. The HAAc-treated sample demonstrated notably reduced values compared to the control, underscoring the efficacy of HAAc treatment in diminishing the existence of oxygen vacancy defects, which are crucial for facilitating electron extraction and transport [27]. To delve deeper into the interaction between HAAc and PbI_2_, Fourier-transform infrared (FTIR) spectrometry was employed, as shown in Appendix A. Compared to the -NH_2_ peaks that were attributed to HA^+^ in the FTIP spectra of HAAc and PbI_2_-HAAc in Figure 2d, it shifted from 3253 cm^−1^ to 3254 cm^−1^, confirming the interaction between HAAc and PbI_2_.

To examine the impact of HAAc treatment on the surface morphology of SnO_2_/ITO films, we utilized atomic force microscopy (AFM) to image both pristine SnO_2_ and HAAc-treated films. As illustrated in Figure 3a,b, the root mean square (RMS) surface roughness of the HAAc-treated film measured a mere 0.663 nm within a 5 × 5 μm test area. In stark contrast, the pristine SnO_2_ displayed an RMS of 0.975 nm. This stark contrast highlights that the HAAc-treated film boasts a notably smoother surface, thereby facilitating the deposition of the perovskite layer with enhanced crystallinity and a reduction in defect density. For a more in-depth exploration of the morphological implications, we employed top-view SEM, as depicted in Figure 3c,e. The HAAc-treated perovskite film prominently featured larger grains in comparison to the control one, as evident in Figure 3d,f. The average grain size for the HAAc-treated film, measuring 797 nm, was significantly greater than the 395 nm observed in the control film. The interaction between perovskite and IL delays the crystallization process of perovskite, resulting in the formation of large-grain crystals in perovskite film. This effectively reduces the number of grain boundaries (GBs), thereby increasing shunt resistance and decreasing series resistance. However, defects tend to form at the grain boundaries, which act as non-radiative recombination centers that diminish device performance and induce perovskite film degradation. Proficient defect passivation by ionic liquids significantly reduced the defect density in perovskite films, thereby decreasing non-radiative recombination in the device and increasing the *V*oc and fill factor. With varying concentrations of HAAc, the improvement of perovskite films is exhibited in different degrees. The mean grain size of the perovskite on 0.2 mg ml^−1^ HAAc is 605 nm, and that on 1 mg ml^−1^ is 548 nm (Appendix A). The notable alterations in morphology are attributed to the enlargement of grain size and the concurrent reduction in grain boundary defects within the perovskite layer, which is an outcome directly resulting from HAAc treatment.

To gauge the impact of our modification methodologies on the perovskite films, we carried out an X-ray diffraction (XRD) analysis. As illustrated in Figure 4a, both the control and HAAc-treated perovskite films displayed an identical diffraction peak at 14.02°, representing the (110) crystal plane of the perovskite. However, the intensity of this (110) peak was significantly enhanced in the HAAc-treated perovskite films. Importantly, no new peaks or peak shifts were detected in the XRD pattern; this observation indicates that the HAAc-treated approach did not alter the orientation of perovskite crystal growth. For a more in-depth exploration of the impact of HAAc on the optical properties of the perovskite films, we conducted steady-state photoluminescence (PL) and time-resolved PL (TRPL) measurements, as illustrated in Figure 4b,c. We utilized test structures consisting of ITO/SnO_2_/perovskite (as the control sample) and ITO/SnO_2_/HAAc/perovskite (as the HAAc-modified sample). While the peak shapes of the control and HAAc-modified films remained largely consistent, the perovskite film treated with HAAc exhibited weakened PL emissions and a shorter decay duration, as detailed in Appendix A [39]. This could be attributed to the modification of the sub-interface, which enhances electron transport. Consequently, electrons are transported through the electron transport layer (ETL) before undergoing radiative recombination, leading to decreased PL intensity and carrier lifetime. These findings provide evidence that HAAc treatment enhances electron transport and reduces non-radiative recombination. Subsequently, we analyzed the UV-visible (UV-vis) absorption spectra of the control and HAAc-treated films, as presented in Figure 4d. Notably, there was almost no change in absorption before and after the modification, indicating that the addition of the ionic liquid HAAc resulted in minimal changes to the optical properties of the perovskite films. To further substantiate the influence of HAAc within the perovskite film in mitigating non-radiative recombination centers at an equivalent biasing voltage (Appendix A), we observed that the electroluminescence (EL) intensity levels of the HAAc-treated device demonstrated a remarkable enhancement in comparison to the control devices. This enhancement signifies a significant reduction in non-radiative recombination, underscoring the effectiveness of the HAAc treatment in improving the electroluminescent characteristics of the device. Furthermore, to gain deeper insights into the impacts of our modification on the surface energy levels of the ETL, we conducted ultraviolet photoelectron spectroscopy (UPS) measurements, as illustrated in Figure 4e. The determination of the work function (WF) and valence band maximum (VBM) was carried out using the following formulas [40]:E_F_ = E_cut-off_ − 21.22
VBM = −E_F_ + E_onset_

The observed values for E_cut-off_ and E_onset_ are 19 and 4.38 eV, respectively, for both the pristine SnO_2_ film and the HAAc-treated SnO_2_ film. Similarly, the VBM for the pristine SnO_2_ and HAAc-treated SnO_2_ films was calculated to be −6.6 and −7.53 eV, respectively. The conduction band minimum (CBM) for the pristine SnO_2_ and HAAc-treated SnO_2_ films was determined to be −2.82 and −3.75 eV, respectively. The energy levels of the perovskite are referenced from pertinent two-step method articles [41]. The energy level details corresponding to this are visually depicted in Figure 4f, while the comprehensive calculation specifics are summarized in Appendix A. These results collectively indicate that the HAAc modification strategy led to smoother electron transport and an increase in electron transport efficiency.

Following this, we performed a quantitative assessment of the trap state density (*n_trap_*) in both the control and HAAc-treated perovskite films within the device architecture, as visually represented in Figure 5a,b. Notably, the HAAc-treated perovskite film demonstrated a notably lower threshold voltage (*V*_TFL_) of 0.382 V in contrast to the control perovskite film, which registered a V_TFL_ of 0.664 V. Furthermore, the *n_trap_* value, calculated utilizing a previously established equation [42], exhibited a significant reduction from 2.1 × 10^15^ cm^−3^ to 1.2 × 10^15^ cm^−3^ in the HAAc-treated perovskite film (as shown in Figure 5c). These observations provide additional evidence for the assertion that the incorporation of HAAc yields effective control over the perovskite film’s morphology. This, in turn, results in a higher-quality film characterized by reduced non-radiative recombination and enhanced carrier transport efficiency. Furthermore, we evaluated carrier transport and recombination within the devices using traditional Mott–Schottky measurements, employing capacitance–voltage analysis. The outcomes revealed a rise in the device’s internal potential (*V*_bi_) from 0.93V to 0.98V upon HAAc modification (Figure 5d), resulting in an enhanced *V*oc owing to improved carrier extraction and transmission rates. To gain a comprehensive understanding of carrier transfer and recombination dynamics within the devices, we conducted electrochemical impedance spectroscopy (EIS) measurements at a bias of 0 V, as depicted in Figure 5e. The EIS curve exhibited two distinct arcs in the high-frequency and low-frequency ranges, corresponding to charge transport resistance (*R*_ct_) and recombination resistance (*R*_rec_), respectively. Additionally, the HAAc-treated device showcased a reduced dark current density, suggesting that the incorporation of HAAc enhanced charge transport performance and minimized the leakage current (Figure 5f).

The photovoltaic performance is an important index to evaluate device performance. Figure 6a showcases the current density–voltage (*J*–*V*) curves of both the control and HAAc-treated PSCs. The control device featuring the bare SnO_2_ substrate achieved a PCE of 20.9%, with a *J*sc of 25.23 mA cm^−2^, a *V*oc of 1.103 V, and an FF of 75.27%. In contrast, the HAAc-treated device emerged as the frontrunner with a champion PCE of 24.16%. It achieved a *J*sc of 25.49 mA cm^−2^, *V*oc of 1.190 V, and FF of 79.63%. It is also noteworthy to mention that Appendix A provides additional evidence of the superior performance exhibited by the HAAc-treated devices. The notable improvements in *V*oc and FF observed in the HAAc-treated devices are primarily attributed to the morphological adjustments that were introduced. This treatment acted as a chemical link connecting SnO_2_ and the perovskite layer, effectively neutralizing surface defects and suppressing the non-radiative recombination. The hysteresis shown by the *J–V* curves for both the control and HAAc-modified devices is depicted in Figure 6b, with the corresponding parameters summarized in Appendix A. Following the modification, there was a notable decrease in the hysteresis index (HI) from 0.117 to 0.046. It is evident that the HAAc-modified device demonstrates minimal hysteresis. This reduction in hysteresis can primarily be ascribed to the decrease in interface defects and the inhibition of perovskite ion migration. The statistical data illustrating the distributions of efficiency for PCEs, presented in Figure 6c, are summarized from at least 30 devices. It not only emphasizes the consistent performance but also highlights the significant differences between HAAc-treated cells and the control devices. The HAAc-treated cells exhibited a tighter distribution and boasted a larger average PCE, exceeding the performance of the control devices. In Figure 6e, the External Quantum Efficiency (EQE) measurements are depicted. The integrated current density values for the control and HAAc-treated cells were calculated to be 24.02 and 24.11 mA cm^−2^, respectively, which is slightly lower than those attained from the *J*–*V* curves. The EQE (External Quantum Efficiency) spectrum of a solar cell (usually ranging from 300 nm to 1100 nm) is integrated over the AM1.5G standard spectrum (according to IEC 60904-3). The EQE spectrum can be converted into the spectral response SR (*λ*), with units of Amp/Watt, while the AM1.5G spectrum is measured in units of Watt/m^2^. Therefore, the integrated unit, Amp/m^2^, represents the current density. The short-circuit current *J*sc from curve IV of the solar cell is obtained by dividing the short-circuit current *J*sc measured under the STC (Standard Test Conditions) by the effective area A of the cell. Hence, the integrated *J*sc from EQE is more independent and reliable [43]. The stable-state maximum power point output, as illustrated in Figure 6d under one sun for a 300 s measurement, reveals a stabilized PCE value of 20.7% for control devices (at 0.82 V) and 23.8% for HAAc-treated cells (at 0.94 V). Furthermore, the investigation encompasses the evaluation of unencapsulated devices utilizing the respective perovskite film under relative humidity conditions of 50% for 1200 h, as depicted in Figure 6e. The HAAc-treated devices demonstrated minimal decay, maintaining an impressive 99.6% of their pristine PCE after the 1200 h period. In contrast, under the same conditions, the PCE of the control devices decreased to 75.1%. In support of this, Appendix A illustrates that the control perovskite film exhibited a significant increase in the PbI_2_ content of XRD spectra, while the HAAc-treated perovskite film showed only a slight increase. These results provide further evidence of the effectiveness of HAAc treatment by improving the stability of the perovskite thin film.

## 3. Conclusions

In conclusion, our work introduced a novel IL into the SnO_2_/perovskite interface, yielding a remarkable optimization of photovoltaic performance in two-step method devices. Through a systematic investigation, we assessed the profound influence of HAAc on the morphology of the perovskite thin film, defect density, and overall photovoltaic performance. Our approach incorporated a range of analytical techniques, including FTIR and XPS analysis, which allowed us to elucidate the significant interactions between the HAAc, perovskite, and SnO_2_, forming a critical foundation for effective defect passivation. Moreover, HAAc played a pivotal role in establishing a chemical bridge between perovskite and SnO_2_, resulting in a greater match of energy level alignment and greatly facilitating electron transport. Leveraging the advantages of effective defect passivation and morphology optimization introduced by HAAc, we successfully mitigated charge carrier recombination, extended PL lifetimes, and achieved a substantial enhancement in device performance. As a result of these innovations, perovskite solar cells treated with ILs achieved an outstanding efficiency of 24.16%, exceeding the efficiency of the control device, which achieved 20.90%. Our comprehensive research not only introduced a new material but also promoted the use of HAAc in two-step photovoltaic devices, setting the stage for improved and more efficient solar energy conversion technologies.

## Figures and Tables

**Figure 1 nanomaterials-14-00653-f001:**
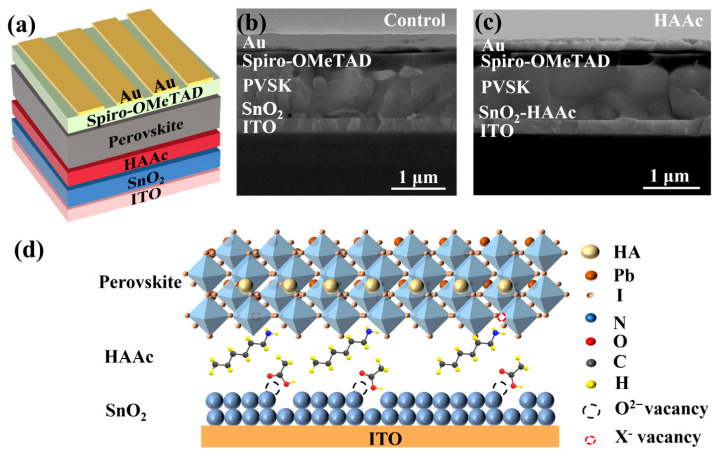
(**a**) Diagrammatic representation of the HAAc-modified PSC structure. (**b**,**c**) Cross-sectional SEM photograph of the perovskite PSC structure. (**d**) Schematic diagram of the formation of the HAAc passivation layer.

**Figure 2 nanomaterials-14-00653-f002:**
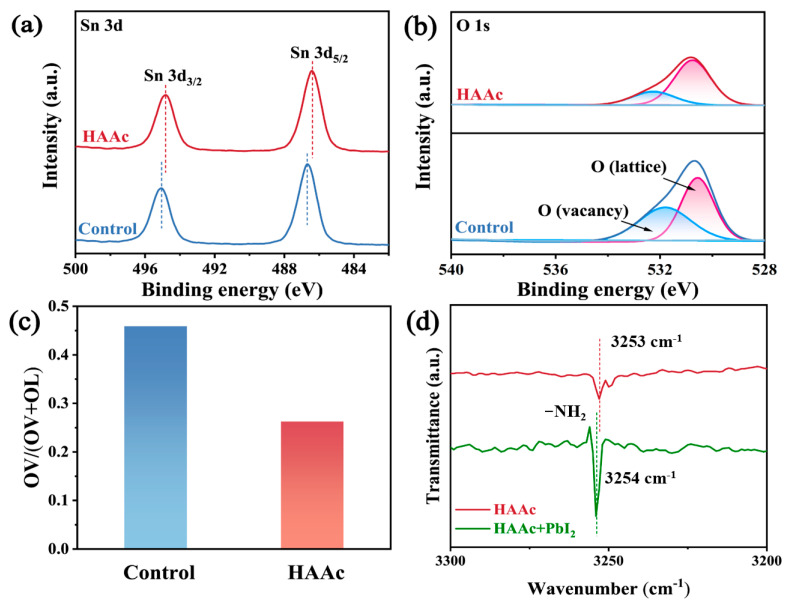
XPS spectra of (**a**) Sn 3d and (**b**) O 1s for SnO_2_ and HAAc-SnO_2_ films. (**c**) Percentage of OV peak area of SnO_2_ and HAAc- SnO_2_. (**d**) The magnified FTIR spectra of HAAc and HAAc-PbI_2_ samples.

**Figure 3 nanomaterials-14-00653-f003:**
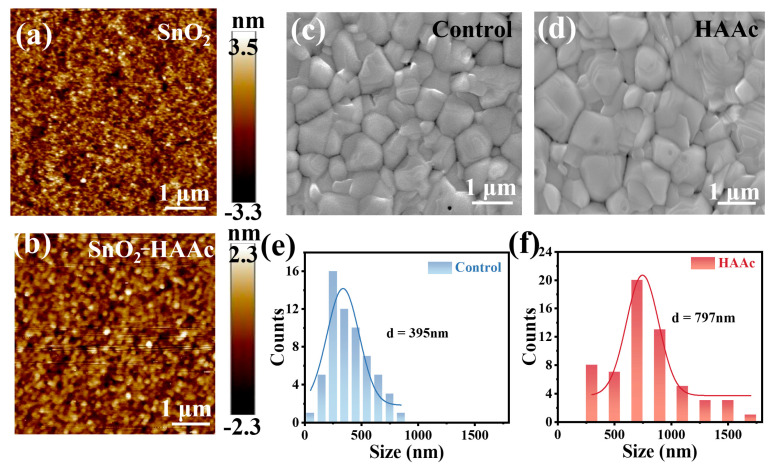
AFM photographs of (**a**) SnO_2_ and (**b**) HAAc-treated SnO_2_ samples. SEM images of (**c**) control and (**d**) HAAc-treated perovskite films. Grain size statistics of (**e**,**f**) perovskite films.

**Figure 4 nanomaterials-14-00653-f004:**
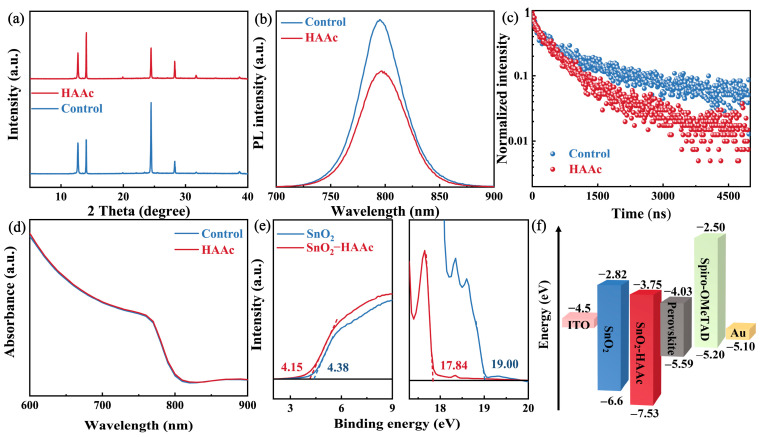
(**a**) XRD spectra of untreated and HAAc-modified perovskite films. (**b**) PL and (**c**) TRPL spectra for the SnO_2_/control and HAAc-modified perovskite samples. (**d**) UV-Vis absorption spectrum of the untreated and HAAc-modified perovskite films. (**e**) UPS tests of SnO_2_ and SnO_2_-HAAc samples. (**f**) Energy level schematic of the components used in this study’s device.

**Figure 5 nanomaterials-14-00653-f005:**
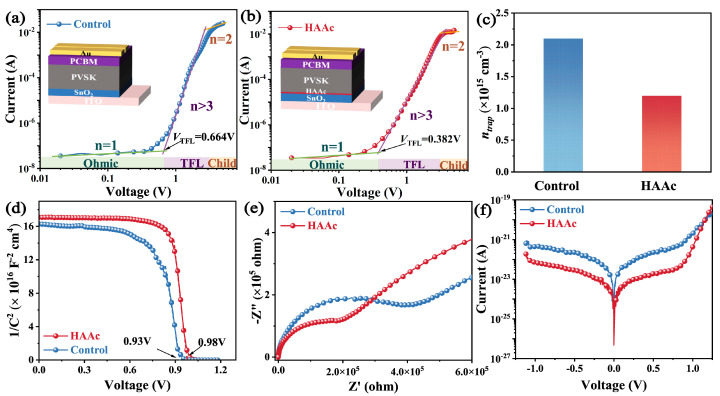
(**a**,**b**) SCLC test of the control and HAAc-modified devices. (**c**) Statistics for computed defect density. (**d**) Mott–Schottky plots of the PSC device. (**e**) Nyquist plots of untreated and HAAc-treated devices. (**f**) Dark *J*–*V* curves of PSC devices.

**Figure 6 nanomaterials-14-00653-f006:**
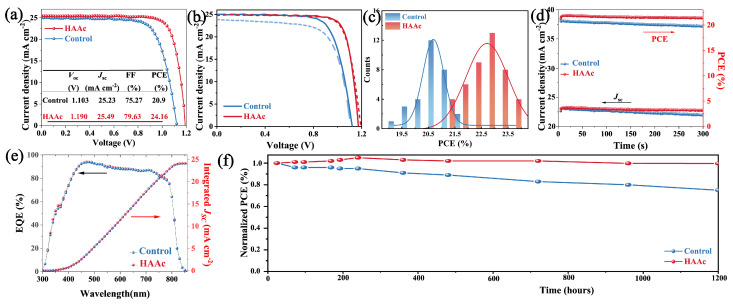
(**a**) *J*–*V* curves of the control and HAAc-modified PSCs. (**b**) *J*–*V* curves showing hysteresis. (**c**) Distribution statistics of control and HAAc-modified device efficiency. (**d**) The stabilized output efficiency of both the control and HAAc-treated devices. (**e**) EQE measurement of HAAc-modified PSCs. (**f**) Stability of control and HAAc-modified devices (under one-sun illumination, without encapsulation, a flowing N_2_ atmosphere, and °C).

## Data Availability

Data are contained within the article and Appendix A.

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
