# Peer review of "Hexylammonium Acetate-Regulated Buried Interface for Efficient and Stable Perovskite Solar Cells"

_nanomaterials, 2024, doi:10.3390/nano14080653_

Round 1

Reviewer 1 Report

Comments and Suggestions for Authors

The authors describe the improvement in perovskite solar cells with the perovskite layer synthesized using the two-step method by introducing an ionic liquid between the SnO2 electron transport layer and the perovskite layer. They report an improvement of almost 20% in performance to 24% efficiency through the improvement in perovskite morphology and defect passivation both in the perovskite layer and at the perovskite/electron transport layer interface. The result in the report are notable and can be published after extensive revision.

Corrections needed:

1)     The fonts of all the figures need to be enlarged much more and thicker lines should be used for better visibility. It is very difficult to read as a printout. 

2)     Please include in Figure 1 the chemical structure of HAAc with lettering rather than a ball and stick image. The ball and stick image is very difficult to see due to its small size in a printout and it is also difficult to associate the atoms with the individual balls.

3)     For Figures 1b and 1c, the contrast of the figures should be aligned such the perovskite layers of the two figures show the matching color palette for better comparison. It looks like even the ITO layer in Figure 1c has bigger grains than the ITO layer in Figure 1b.

4)     In the first paragraph of the Results and Discussion, please include a few sentences describing the two step method and contrast it with the single step method.

5)     For the FTIR image in Figure 2d, please indicate the -NH2 peaks and show the peak shifts clearly.

6)     I am confused by the PL measurements in Figure 4 and S7; I do not understand why the SnO2 data is placed in the Supporting Information while the glass data is in the main manuscript. I don’t what is the point of the glass measurement, what is it’s purpose? The SnO2 data is pretty clear on its own. I suggest removing the glass data from the manuscript completely.

7)     In figure 4e, the energy diagram is very small when printed out so it is difficult to read. I suggest expanding that image and instead use a simple line image rather than a complicated block image so that you can use larger fonts.

8)     The authors should also include a discussion how the improvement morphology and defect passivation each separately improve the performance of the solar cell. For example, the increase in grain size of the perovskite probably improves current transport by increasing shunt resistance and decreasing series resistance, but probably has little effect on Voc.

9)     I have certain reservations about the reported efficiencies, based on the fill-factor. Typically solar cells with such high efficiencies above 20% have fill factors exceeding 80%. I would suggest the authors to revisit their estimations of the solar cell area when calculating the short circuit current flux.

Comments on the Quality of English Language

Editing corrections

Line 35: “ETLs, TiO2” instead of “ETL, ”

Line 44: “understanding” instead of “comprehension”; “the correlations between the” instead of “the correlations the”.

Abstract correction:

Due to current issues of energy level mismatch and low transport efficiency in commonly used electron transport layers (ETLs), such as TiOand SnO2, finding a more effective method to passivate the ETL and perovskite interface has become an urgent matter. In this work, we have integrated a new material, the ionic liquid (IL) hexylammonium acetate (HAAc) into the SnO2/perovskite interface to improve performance via improvenement of perovskite quality formed by the two-step method. The IL anions fill oxygen vacancy defects in SnO2, while the IL cations interact chemically with Pb2+ within the perovskite structure reducing defects and optimizing the morphology of the perovskite film such that the energy levels of the ETL and perovskite become better matched. Consequently, the decrease in nonradiative recombination promotes enhanced electron transport efficiency. Utilizing HAAc, we successfully regulated the morphology and defect states of the perovskite layer, resulting in devices surpassing 24% efficiency. This research breakthrough not only introduces a novel material but also propels the utilization of ILs in enhancing the performance of perovskite photovoltaic systems using two-step synthesis.

Author Response

Response to Reviewer’s Comments (Manuscript ID: nanomaterials-2955195)

Manuscript title: Hexylammonium Acetate-Regulated Buried Interface for Efficient and Stable Perovskite Solar Cells

Authors: Ruiyuan Hu*, Taomiao Wang, Fei Wang, Yongjun Li, Yonggui Sun, Xiao Liang, Xianfang Zhou, Guo Yang, Qiannan Li, Fan Zhang, Quanyao Zhu, Xing’ao Li*, Hanlin Hu*

Dear Editor and Reviewers,

Thank you very much for your kind consideration for giving us the opportunity to revise our article. We deeply appreciate the constructive comments from the reviewer which we believe will help improve our manuscript. We have considered all of the reviewers’ comments and revised the manuscript accordingly. For convenience, in the PDF version of this letter, we outline the reviewer’ comments with black font, our responses to the specific comments with blue font, and changes made to the manuscript with red font.

Thank you very much for your time, and we look forward to your positive response.

Best wishes

Hanlin Hu

*********************************************************************************************************************

Reviewer: 1

Comments to the Author

The authors describe the improvement in perovskite solar cells with the perovskite layer synthesized using the two-step method by introducing an ionic liquid between the SnO2 electron transport layer and the perovskite layer. They report an improvement of almost 20% in performance to 24% efficiency through the improvement in perovskite morphology and defect passivation both in the perovskite layer and at the perovskite/electron transport layer interface. The result in the report are notable and can be published after extensive revision.

Response: We like to thank the reviewer for the constructive comments.

1) The fonts of all the figures need to be enlarged much more and thicker lines should be used for better visibility. It is very difficult to read as a printout. 

Response: We like to thank the reviewer for the careful check. We redrew all the figures with thicker lines and larger fonts as shown in the updated manuscript.

Action: (In manuscript,)

Figure 1. (a) The diagrammatic representation of the HAAc-modified PSCs structure. (b,c) Cross-sectional SEM photograph of the perovskite PSCs structure. (d) Schematic diagram of the formation of the HAAc passivation layer.

Figure 2. XPS spectra of (a) Sn 3d and (b) O 1s for SnO2 and HAAc-SnO2 films. (c) Percentage of OV peak area of SnO2 and HAAc- SnO2. (d) The magnified FTIR spectra of HAAc and HAAc-PbI2 samples.

Figure 3. AFM photographs of (a) SnO2 and (b) HAAc-treated SnO2 samples. SEM images of (c) control and (e) HAAc-treated perovskite films. Grain size statistics of (e, f) perovskite films.

Figure 4. (a) XRD spectra of untreated and HAAc-modified perovskite films. (b) PL and (c) TRPL spectra for the SnO2/control and HAAc-modified perovskite samples. (d) UV-Vis absorption spectrum of the untreated and HAAc-modified perovskite films. (e) UPS tests of SnO2 and SnO2-HAAc samples. (f) Energy level schematic of the components used in this study's device.

Figure 5. (a, b) SCLC test of the control and HAAc-modified devices. (c) Statistics for computed defect density. (d) Mott–Schottky plots of PSCs device. (e) Nyquist plots of untreated and HAAc-treated devices. (f) Dark J–V curves of PSCs devices.

Figure 6. (a) J-V curves of the control and HAAc-modified PSCs. (b) J-V curves hysteresis. (c) Distribution statistics of control and HAAc-modified device efficiency. (d) The stabilized output efficiency of both the control and HAAc-treated devices. (e) EQE measurement of HAAc-modified PSCs. (f) Stability of control and HAAc-modified devices (under one-sun illumination, without encapsulation, flowing N2 atmosphere, 40℃).

2) Please include in Figure 1 the chemical structure of HAAc with lettering rather than a ball and stick image. The ball and stick image is very difficult to see due to its small size in a printout and it is also difficult to associate the atoms with the individual balls.

Response: We like to thank the reviewer for the valuable suggestion. We have made modifications to the ball-and-stick model of HAAc in Figure 1 to make it easier to observe clearly.

Action: (In manuscript, page 4)

Figure 1. (a) The diagrammatic representation of the HAAc-modified PSCs structure. (b,c) Cross-sectional SEM photograph of the perovskite PSCs structure. (d) Schematic diagram of the formation of the HAAc passivation layer.

3) For Figures 1b and 1c, the contrast of the figures should be aligned such the perovskite layers of the two figures show the matching color palette for better comparison. It looks like even the ITO layer in Figure 1c has bigger grains than the ITO layer in Figure 1b.

Response: We like to thank the reviewer for the careful check. We use the same batch of ITO glasses. The contrast of the figures was aligned. The related image has been updated as shown in the revision.

Action: (In manuscript, page 3)

Figure 1. (a) The diagrammatic representation of the HAAc-modified PSCs structure. (b, c) Cross-sectional SEM photograph of the perovskite PSCs structure. (d) Schematic diagram of the formation of the HAAc passivation layer.

4) In the first paragraph of the Results and Discussion, please include a few sentences describing the two step method and contrast it with the single step method.

Response: We like to express our gratitude to the reviewer for the valuable advice. Moreover, we have incorporated an in-depth comparison and analysis between the one-step and two-step methods in the Introduction Part as shown in the revised version.

Action: (In manuscript, page 2)

Owing to the advantages demonstrated by ILs in optimizing the hole transport layer (HTL)/perovskite interface, the search for highly efficient and practical ILs has now emerged as a foremost priority in this field. For the current mainstream preparation methods of perovskite, the one-step method has proven to be efficient in producing smooth and compact perovskite films. However, the antisolvent process, while effective, is difficult to control and unsuitable for fabricating large-area devices, often results in poor surface coverage and diminished cell performance[36,37]. In contrast, the utilization of a two-step deposition method offers a more effective and manageable solution. This method involves the separate deposition of the organic layer and PbI2 layer.

5) For the FTIR image in Figure 2d, please indicate the -NH2 peaks and show the peak shifts clearly.

Response: We like to thank the reviewer for your careful check. The -NH2 peaks have been clearly indicated in the FTIR image as suggested.

Action: (In manuscript, page 5)

Figure 2. XPS spectra of (a) Sn 3d and (b) O 1s for SnO2 and HAAc-SnO2 films. (c) Percentage of OV peak area of SnO2 and HAAc- SnO2. (d) The magnified FTIR spectra of HAAc and HAAc-PbI2 samples.

6) I am confused by the PL measurements in Figure 4 and S7; I do not understand why the SnO2 data is placed in the Supporting Information while the glass data is in the main manuscript. I don’t what is the point of the glass measurement, what is it’s purpose? The SnO2 data is pretty clear on its own. I suggest removing the glass data from the manuscript completely.

Response: We like to thank the reviewer for the valuable comments. We displayed the PL spectra on SnO2 in the main manuscript and removed the glass data as shown in the revised version.

Action: (In manuscript, page 7)

Figure 4. (a) XRD spectra of untreated and HAAc-modified perovskite films. (b) PL and (c) TRPL spectra for the SnO2/control and HAAc-modified perovskite samples. (d) UV-Vis absorption spec-trum of the untreated and HAAc-modified perovskite films. (e) UPS tests of SnO2 and SnO2-HAAc samples. (f) Energy level schematic of the components used in this study's device.

7) In figure 4e, the energy diagram is very small when printed out so it is difficult to read. I suggest expanding that image and instead use a simple line image rather than a complicated block image so that you can use larger fonts.

Response: We like to appreciate the reviewer for this valuable comment. As suggested, we redrew the energy diagram with larger fonts.

Action: (In manuscript, page 7)

Figure 4. (a) XRD spectra of untreated and HAAc-modified perovskite films. (b) PL and (c) TRPL spectra for the SnO2/control and HAAc-modified perovskite samples. (d) UV-Vis absorption spectrum of the untreated and HAAc-modified perovskite films. (e) UPS tests of SnO2 and SnO2-HAAc samples. (f) Energy level schematic of the components used in this study's device. 

8)  The authors should also include a discussion how the improvement morphology and defect passivation each separately improve the performance of the solar cell. For example, the increase in grain size of the perovskite probably improves current transport by increasing shunt resistance and decreasing series resistance, but probably has little effect on Voc.

Response: We like to appreciate the reviewer for this valuable comment and insightful suggestions. Following the interface modification by ILs, we achieved a superior-quality perovskite thin film with increased grain sizes, resulting in a notable decrease in grain boundaries, increased shunt resistance and decreased series resistance. This enhancement proves advantageous for optimizing current transport within the device. The revised version comprehensively discussed the detailed examination of the effects of defect passivation on the device. Due to the proficient defect passivation facilitated by ionic liquids, there is a marked reduction in the defect density of perovskite thin films, leading to diminished non-radiative recombination and improved Voc and fill factor in the device.

Action: (In manuscript, page 5)

The average grain size for the HAAc-treated film, measuring 797 nm, is significantly greater than the 395 nm observed in the control film. The interaction between perovskite and IL delays the crystallization process of perovskite, resulting in the formation of large-grain crystals in perovskite film. This effectively reduces the number of grain boundaries (GBs), thereby increasing shunt resistance and decreasing series resistance. However, defects tend to form at grain boundaries, acting non-radiative recombination centers that diminish device performance and induce perovskite film degradation. Proficient defect passivation by ionic liquids significantly reduced the defect density in perovskite films, thereby decreasing non-radiative recombination in the device and increased Voc and fill factor. With varying concentrations of HAAc, the improvement of perovskite films exhibits in different degrees.

9) I have certain reservations about the reported efficiencies, based on the fill-factor. Typically solar cells with such high efficiencies above 20% have fill factors exceeding 80%. I would suggest the authors to revisit their estimations of the solar cell area when calculating the short circuit current flux.

Response: We like to thank the reviewer for pointing this out. The fill factor of perovskite solar cells reflects a comprehensive metric influenced by various factors. To validate the reliability and consistency of our efficiency measurements, we conducted statistical analyses on multiple batches of devices, examining their photovoltaic parameters (Table R1). Our results revealed that devices treated with ionic liquids demonstrated exceptional Voc, contributing to their exceptional photovoltaic performance. Moreover, we meticulously calibrated the effective area of the devices using a mask, resulting in a finalized device active area of 0.04 cm². Additionally, we compiled pertinent research pertaining to high power conversion efficiencies and fill factors below 80% (Table R2).

Table R1. The multiple sets of photovoltaic parameters

Jsc

(mA cm-2)

Voc(V)

FF

PCE (%)

1

25.61

1.188

78.05

23.74

2

25.34

1.182

79.36

23.88

3

25.21

1.188

78.29

23.57

4

25.45

1.185

77.88

23.59

5

25.69

1.188

76.90

23.59

6

25.43

1.186

78.38

23.69

7

25.27

1.190

78.70

23.70

8

25.44

1.194

77.65

23.61

9

25.03

1.185

79.40

23.73

10

25.24

1.184

78.35

23.58

Table R2. The Research related to high power conversion efficiency and fill factor below 80%.

Device Structure

Jsc

(mA cm-2)

Voc(V)

FF

PCE(%)

ref.

ITO/SnO2/Cs0.03 (FA0.97 MA0.03 )0.97 Pb(I0.97 Br0.03 )3/ spiro-OMeTAD /

24.57

1.21

80.00

23.78

[1]

FTO/compact-TiO2/thin-mesoporous TiO2/Cs0.06FA0.79MA0.15Pb(I0.85Br0.15)3/spiro-OMeTAD /Au

23.2

1.17

79.00

21.50

[2]

FTO/SnO2/CH3NH3PbI3/spiro-OMeTAD/Au

22.42

1.18

77.00

20.40

[3]

ITO/SnO2/ Cs0.05(FA0.95MA0.05)0.95Pb(I0.95Br0.05)3/spiro-OMeTAD/Au

25.17

1.18

79.00

23.56

[4]

ITO/SnO2 /perovskite/PEAI/spiro-OMeTAD/Au

25.2

1.179

78.40

23.32

[5]

FTO/compact-TiO2/thin-mesoporous TiO2/ (FAPbI3)0.92(MAPbBr3)0.08/spiro-OMeTAD /Au

24.2

1.19

78.50

22.60

[6]

ITO/SnO2 /(FAPbI3)0.96(MAPbBr3)0.04/ spiro-OMeTAD/Ag

24.91

1.17

79.50

23.22

[7]

ITO/SnO2 /perovskite/spiro-OMeTAD/Ag

25.47

1.19

79.32

24.04

[8]

ITO/SnO2/NSE/ Cs0.05FA0.81MA0.14PbI20.55Br0.45/Spiro-OMeTAD/Au

22.84

1.19

77.93

21.18

[9]

ITO/SnO2 / Cs0.1(FA0.83MA0.17)0.9Pb(I0.83Br0.17)3 /MoO3/Ag

23.66

1.19

75.00

21.12

[10]

References

  1. Yang, G.; Ren, Z.; Liu, K.; Qin, M.; Deng, W.; Zhang, H.; Wang, H.; Liang, J.; Ye, F.; Liang, Q.; et al. Stable and Low-Photovoltage-Loss Perovskite Solar Cells by Multifunctional Passivation. Nat. Photonics 2021, 15, 681–689, doi:10.1038/s41566-021-00829-4.
  2. Abdi-Jalebi, M.; Andaji-Garmaroudi, Z.; Cacovich, S.; Stavrakas, C.; Philippe, B.; Richter, J.M.; Alsari, M.; Booker, E.P.; Hutter, E.M.; Pearson, A.J.; et al. Maximizing and Stabilizing Luminescence from Halide Perovskites with Potassium Passivation. Nature 2018, 555, 497–501, doi:10.1038/nature25989.
  3. Wang, Y.; Liang, Y.; Zhang, Y.; Yang, W.; Sun, L.; Xu, D. Pushing the Envelope: Achieving an Open-Circuit Voltage of 1.18 V for Unalloyed MAPbI3 Perovskite Solar Cells of a Planar Architecture. Adv. Funct. Mater. 2018, 28, 1–8, doi:10.1002/adfm.201801237.
  4. Yang, S.; Han, Q.; Wang, L.; Zhou, Y.; Yu, F.; Li, C.; Cai, X.; Gao, L.; Zhang, C.; Ma, T. Over 23% Power Conversion Efficiency of Planar Perovskite Solar Cells via Bulk Heterojunction Design. Chem. Eng. J. 2021, 426, 131838, doi:10.1016/j.cej.2021.131838.
  5. Jiang, Q.; Zhao, Y.; Zhang, X.; Yang, X.; Chen, Y.; Chu, Z.; Ye, Q.; Li, X.; Yin, Z.; You, J. Surface Passivation of Perovskite Film for Efficient Solar Cells. Nat. Photonics 2019, 13, 460–466, doi:10.1038/s41566-019-0398-2.
  6. Yoo, J.J.; Wieghold, S.; Sponseller, M.C.; Chua, M.R.; Bertram, S.N.; Hartono, N.T.P.; Tresback, J.S.; Hansen, E.C.; Correa-Baena, J.P.; Bulović, V.; et al. An Interface Stabilized Perovskite Solar Cell with High Stabilized Efficiency and Low Voltage Loss. Energy Environ. Sci. 2019, 12, 2192–2199, doi:10.1039/c9ee00751b.
  7. Yang, I.S.; Park, N.G. Dual Additive for Simultaneous Improvement of Photovoltaic Performance and Stability of Perovskite Solar Cell. Adv. Funct. Mater. 2021, 31, 1–7, doi:10.1002/adfm.202100396.
  8. He, J.; Sheng, W.; Yang, J.; Zhong, Y.; Cai, Q.; Liu, Y.; Guo, Z.; Tan, L.; Chen, Y. Synchronous Elimination of Excess Photoinstable PbI 2 and Interfacial Band Mismatch for Efficient and Stable Perovskite Solar Cells . Angew. Chemie 2024, 136, doi:10.1002/ange.202315233.
  9. Zheng, D.; Peng, R.; Wang, G.; Logsdon, J.L.; Wang, B.; Hu, X.; Chen, Y.; Dravid, V.P.; Wasielewski, M.R.; Yu, J.; et al. Simultaneous Bottom-Up Interfacial and Bulk Defect Passivation in Highly Efficient Planar Perovskite Solar Cells Using Nonconjugated Small-Molecule Electrolytes. Adv. Mater. 2019, 31, 1–9, doi:10.1002/adma.201903239.
  10. Wu, S.; Li, Z.; Zhang, J.; Liu, T.; Zhu, Z.; Jen, A.K.Y. Efficient Large Guanidinium Mixed Perovskite Solar Cells with Enhanced Photovoltage and Low Energy Losses. Chem. Commun. 2019, 55, 4315–4318, doi:10.1039/c9cc00016j.

Editing corrections

1) Line 35: “ETLs, TiO2” instead of “ETL,”

Response: We are grateful for this careful check! We have rectified it accordingly as shown in the revised manuscript.

Action: (In manuscript, page 1)

Among the commonly investigated ETL, TiO2 has played a pioneering and thoroughly researched role in the development of PSCs[5,6]. However, TiO2 is not without its limitations.

2) Line 44: “understanding” instead of “comprehension”; “the correlations between the” instead of “the correlations the”.

Response: We like to thank reviewer for the comprehensive review. We have corrected it accordingly as suggested as shown in the revision.

Action: (In manuscript, page 1)

Establishing a understanding comprehension of the correlations the characteristics of SnO2 and the performance exhibited by PSCs is essential to fully exploit its potential as an ETL[12–14]

3) Abstract correction:

 Due to current issues of energy level mismatch and low transport efficiency in commonly used electron transport layers (ETLs), such as TiO2 and SnO2, finding a more effective method to passivate the ETL and perovskite interface has become an urgent matter. In this work, we have integrated a new material, the ionic liquid (IL) hexylammonium acetate (HAAc) into the SnO2/perovskite interface to improve performance via improvenement of perovskite quality formed by the two-step method. The IL anions fill oxygen vacancy defects in SnO2, while the IL cations interact chemically with Pb2+ within the perovskite structure reducing defects and optimizing the morphology of the perovskite film such that the energy levels of the ETL and perovskite become better matched. Consequently, the decrease in nonradiative recombination promotes enhanced electron transport efficiency. Utilizing HAAc, we successfully regulated the morphology and defect states of the perovskite layer, resulting in devices surpassing 24% efficiency. This research breakthrough not only introduces a novel material but also propels the utilization of ILs in enhancing the performance of perovskite photovoltaic systems using two-step synthesis.

Response: We like to appreciate reviewer for this useful feedback! We have made the corrections   accordingly as suggested.

Action: (In manuscript, page 1)

Due to current issues of energy level mismatch and low transport efficiency in commonly used electron transport layers (ETLs), such as TiO2 and SnO2, finding a more effective method to passivate the ETL and perovskite interface has become an urgent matter. In this work, we have integrated a new material, the ionic liquid (IL) hexylammonium acetate (HAAc) into the SnO2/perovskite interface to improve performance via improvement of perovskite quality formed by the two-step method. The IL anions fill oxygen vacancy defects in SnO2, while the IL cations interact chemically with Pb2+ within the perovskite structure reducing defects and optimizing the morphology of the perovskite film such that the energy levels of the ETL and perovskite become better matched. Consequently, the decrease in nonradiative recombination promotes enhanced electron transport efficiency. Utilizing HAAc, we successfully regulated the morphology and defect states of the perovskite layer, resulting in devices surpassing 24% efficiency. This research breakthrough not only introduces a novel material but also propels the utilization of ILs in enhancing the performance of perovskite photovoltaic systems using two-step synthesis.

Reviewer 2 Report

Comments and Suggestions for Authors

The work reports is quite nice and complete.  The English in this manuscript should be checked and revised carefully.

Line 140  Should be Figure 2 a)

Line 255  “The photovoltaic performance is an important index to evaluate battery performance.”  This is not a battery?

Comments on the Quality of English Language

There are some words that are not correct and some sentences that need to be adjusted.

Author Response

Response to Reviewer’s Comments (Manuscript ID: nanomaterials-2955195)

Manuscript title: Hexylammonium Acetate-Regulated Buried Interface for Efficient and Stable Perovskite Solar Cells

Authors: Ruiyuan Hu*, Taomiao Wang, Fei Wang, Yongjun Li, Yonggui Sun, Xiao Liang, Xianfang Zhou, Guo Yang, Qiannan Li, Fan Zhang, Quanyao Zhu, Xing’ao Li*, Hanlin Hu*

Dear Editor and Reviewers,

Thank you very much for your kind consideration for giving us the opportunity to revise our article. We deeply appreciate the constructive comments from the reviewer which we believe will help improve our manuscript. We have considered all of the reviewers’ comments and revised the manuscript accordingly. For convenience, in the PDF version of this letter, we outline the reviewer’ comments with black font, our responses to the specific comments with blue font, and changes made to the manuscript with red font.

Thank you very much for your time, and we look forward to your positive response.

Best wishes

Hanlin Hu

*********************************************************************************************************************

Reviewer: 2

The work reports is quite nice and complete.  The English in this manuscript should be checked and revised carefully.

Response: We like to thank reviewer for this kind comment. In response, we have invited Prof. Annie NG, an English-native speaker from the Department of Electrical and Computer Engineering at Nazarbayev University in Kazakhstan, to refine the language in this manuscript.  

1) Line 140  Should be Figure 2 a)

Response: We like to thank reviewer for the careful check! We have corrected it accordingly.

Action: (In manuscript, page 7)

The chemical interplay between HAAc and SnO2 is vividly illustrated through the utilization of X-ray photoelectron spectroscopy (XPS), the whole surveys of which are displayed in Figure S4. In Figure 2(a), a conspicuous shift is observed in both the Sn 3d5/2 and Sn 3d3/2 peaks within the SnO2, transitioning from 495.1 eV and 486.7 eV to 494.8 eV and 486.4 eV, respectively.

2) Line 255  “The photovoltaic performance is an important index to evaluate battery performance.”  This is not a battery?

Response: We like to thank reviewer for pointing this out. We have corrected it as suggested.

Action: (In manuscript, page 8)

The photovoltaic performance is an important index to evaluate device performance. Figure 6(a) showcases the current density-voltage (J-V) curves of both the control and HAAc-treated PSCs.

Reviewer 3 Report

Comments and Suggestions for Authors

I have thoroughly reviewed your manuscript titled "Hexylammonium Acetate-Regulated Buried Interface for Efficient and Stable Perovskite Solar Cells" and I am pleased to provide my evaluation. The study presents an intriguing approach to improving the performance of perovskite solar cells (PSCs) using hexylammonium acetate (HAAc) for interface engineering. The results indicating an increase in efficiency and stability are particularly promising.

However, to strengthen the manuscript and validate the findings, I recommend addressing the following points:

1.       Chemicals in the same series as HAAc (hexylammonium acetate) include Methylammonium acetate (MAAc), Ethylammonium acetate (EAAc), Propylammonium acetate (PAAc), Butylammonium acetate (BAAc), and Pentylammonium acetate (PeAAc). Would these have the same effect? Why?

2.       Does the treatment with different concentrations of HAAc effectively reduce the RMS in all cases?

3.       Could the author elaborate on the film structure used in Figure 4(b) for the photoluminescence (PL) and (c) for the time-resolved photoluminescence (TRPL) measurements?

4.       I recommend that you read the paper thoroughly at https://doi.org/10.1002/adfm.202306367 for an in-depth analysis and explanation of photoluminescence (PL) and time-resolved photoluminescence (TRPL) related to the Electron Transport Layer (ETL) and Hole Transport Layer (HTL). After reviewing, please incorporate the relevant discussion into your study.

5.       In Figure 6(a), which shows the photovoltaic parameters and J-V curves of the control and HAAc-modified perovskite solar cells, the author should also provide data for both forward and reverse scans, as well as the hysteresis coefficient. Does the modification with HAAc improve the hysteresis behavior of the cells?

6.       In Figure 6(c), it is mentioned that "The cumulative current density values for the HAAc-treated cells were calculated to be 24.11 mA cm-2, aligning well with the values derived from the J-V curves." The author should also present the External Quantum Efficiency (EQE) results and the cumulative current density values for the control group, and discuss whether these are consistent with the J-V results.

7.       The author needs to clearly state the number of cells tested in Figure 6(e) and also include the standard deviation for the data presented. If only one cell each was tested for the control and HAAc-modified devices, is that representative enough to validate the results?

Author Response

ccResponse to Reviewer’s Comments (Manuscript ID: nanomaterials-2955195)

Manuscript title: Hexylammonium Acetate-Regulated Buried Interface for Efficient and Stable Perovskite Solar Cells

Authors: Ruiyuan Hu*, Taomiao Wang, Fei Wang, Yongjun Li, Yonggui Sun, Xiao Liang, Xianfang Zhou, Guo Yang, Qiannan Li, Fan Zhang, Quanyao Zhu, Xing’ao Li*, Hanlin Hu*

Dear Editor and Reviewers,

Thank you very much for your kind consideration for giving us the opportunity to revise our article. We deeply appreciate the constructive comments from the reviewer which we believe will help improve our manuscript. We have considered all of the reviewers’ comments and revised the manuscript accordingly. For convenience, in the PDF version of this letter, we outline the reviewer’ comments with black font, our responses to the specific comments with blue font, and changes made to the manuscript with red font.

Thank you very much for your time, and we look forward to your positive response.

Best wishes

Hanlin Hu

*********************************************************************************************************************

Reviewer: 3

I have thoroughly reviewed your manuscript titled "Hexylammonium Acetate-Regulated Buried Interface for Efficient and Stable Perovskite Solar Cells" and I am pleased to provide my evaluation. The study presents an intriguing approach to improving the performance of perovskite solar cells (PSCs) using hexylammonium acetate (HAAc) for interface engineering. The results indicating an increase in efficiency and stability are particularly promising.

However, to strengthen the manuscript and validate the findings, I recommend addressing the following points:

1) Chemicals in the same series as HAAc (hexylammonium acetate) include Methylammonium acetate (MAAc), Ethylammonium acetate (EAAc), Propylammonium acetate (PAAc), Butylammonium acetate (BAAc), and Pentylammonium acetate (PeAAc). Would these have the same effect? Why?

Response: We like to thank the reviewer for the valuable comment and suggestions. In response, we have designed an experiment focusing on acetate-based ionic liquid-modified perovskite solar cells. Analysis of the photoluminescence (PL) of SnO2/PVSK films (Figure R1) reveals a decrease in all PL peaks of IL-modified decrease compared to the control film. This can be understood as these acetate-based ionic liquids' positive effects on interface modification stem from passivating interface defects and adjusting work functions, enhancing the electron transport capability of tin dioxide. Regarding the specific comparative work on this series of ionic liquids, we will continue to focus on their modulation of work functions, the strength of chemical interactions, tin dioxide surface morphology, and the regulation of perovskite crystallization processes in subsequent work, aiming for a comprehensive exploration and understanding. We look forward to further elucidating their applications in subsequent work.

Figure R1. The PL spectra of perovskite films modified by different ILs.

2) Does the treatment with different concentrations of HAAc effectively reduce the RMS in all cases?

Response: We would like to thank the reviewer for pointing this out. We have supplemented the AFM images of tin dioxide modified with ionic liquids at different concentrations and found that the surface roughness of the SnO2 films modified with ionic liquids at four concentrations (0.25/0.5/1/1.5 mg/mL) was reduced to varying degrees in comparison to the reference SnO2 films (Figure R2). This phenomenon validates the effective improvement of surface roughness by ionic liquids on tin dioxide. In addition, the optimal concentration of 0.5 mg/mL resulted in the lowest roughness, which is also consistent with the optimal concentration used for device modification. To ensure result matching and obtain clearer AFM images, we updated the original AFM images and conducted multiple repeated experiments to verify the reliability of their morphology (Figure R3).

Figure R2. AFM images of tin dioxide modified with ionic liquids at different concentrations.

Figure R3. AFM images of tin dioxide modified with ionic liquids.

Action: (In manuscript, page 6)

As illustrated in Figure 3(a) and (b), the root mean square (RMS) surface roughness of the HAAc-treated film measured a mere 0.663 nm within a 5×5 μm test area. In stark contrast, the pristine SnO2 displayed an RMS of 0.975 nm. This stark contrast highlights that the HAAc-treated film boasts a notably smoother surface, thereby facilitating the deposition of the perovskite layer with enhanced crystallinity and reduction in defect density.

3) Could the author elaborate on the film structure used in Figure 4(b) for the photoluminescence (PL) and (c) for the time-resolved photoluminescence (TRPL) measurements?

Response: We like to appreciate the valuable suggestion from the reviewer.  The film structure of the samples for PL and TRPL is ITO/SnO2/perovskite (control sample) and ITO/SnO2/HAAc/perovskite (HAAc sample)

Action: (In manuscript, page 6)

For a more in-depth exploration of the impact of HAAc on the optical properties of the perovskite films, we conducted steady-state photoluminescence (PL) and time-resolved PL (TRPL) measurements, as illustrated in Figure 4(b) and (c). We utilized structures we used were ITO/SnO2/perovskite (control sample) and ITO/SnO2/HAAc/perovskite (HAAc-modified sample). While the peak shapes of the control and HAAc-modified films remained largely consistent, the perovskite film treated with HAAc exhibited weakened PL emission and shorter decay lifetimes, as detailed in Table S2 [40].

4) I recommend that you read the paper thoroughly at https://doi.org/10.1002/adfm.202306367 for an in-depth analysis and explanation of photoluminescence (PL) and time-resolved photoluminescence (TRPL) related to the Electron Transport Layer (ETL) and Hole Transport Layer (HTL). After reviewing, please incorporate the relevant discussion into your study.

Response: We like to thank the reviewer for his/her valuable suggestion. We’ve learnt the paper about the in-depth analysis and explanation of PL and TRPL, and modified the explanation in the manuscript.

Action: (In manuscript, page 6)

For a more in-depth exploration of the impact of HAAc on the optical properties of the perovskite films, we conducted steady-state photoluminescence (PL) and time-resolved PL (TRPL) measurements, as illustrated in Figure 4(b) and (c). We utilized test structures consisting ITO/SnO2/perovskite (control sample) and ITO/SnO2/HAAc/perovskite (HAAc-modified sample). While the peak shapes of the control and HAAc-modified films remained largely consistent, the perovskite film treated with HAAc exhibited weakened PL emission and shorter decay lifetimes, as detailed in Table S2 [40]. This can be  attributed to the modification of the sub-interface, which enhances electron transport. Consequently, electrons are transported through the electron transport layer (ETL) before undergoing radiative recombination, leading to decreased PL intensity and carrier lifetime. These findings provide evidence that HAAc treatment enhances electron transport and reduces non-radiative recombination. Subsequently, we analyzed the UV-visible (UV-vis) absorption spectra of the control and HAAc-treated films, as presented in Figure 4d.

40 Lee, K.; Huang, Y.; Chiu, W.; Huang, Y.; Chen, G.; Adugna, G.B.; Li, S.; Lin, F.; Lu, S.; Hsieh, H. Fluorinated Pentafulvalene-Fused Hole-Transporting Material Enhances the Performance of Perovskite Solar Cells with Efficiency Exceeding 23 %. Adv. Funct. Mater. 2023, 2306367, 1–11, doi:10.1002/adfm.202306367.

Action: (In Supporting Information, page 6)

Table S2. Summary of the fitting outcomes and associated dynamic parameters extracted from TRPL decay traces.

Sample

A1 (%)

τ1 (ns)

A2 (%)

τ2 (ns)

τavg. (ns)

Control

0.39

122.55

0.36

1028.69

925.11

HAAc-modified

0.43

83.11

0.48

658.54

591.65

5) In Figure 6(a), which shows the photovoltaic parameters and J-V curves of the control and HAAc-modified perovskite solar cells, the author should also provide data for both forward and reverse scans, as well as the hysteresis coefficient. Does the modification with HAAc improve the hysteresis behavior of the cells?

Response: We thank the reviewer for his/her valuable suggestion. We have added the hysteresis data in revised version, and the parameters are summarized in Table S3. After modification, the hysteresis index (HI) decreased from 0.117 to 0.046. It can be observed that the HAAc-modified device exhibits negligible hysteresis. The reduction in hysteresis can be mainly attributed to the decrease in interface defects and the inhibition of perovskite ion migration.

Action: (In manuscript, page 8,9)

Figure 6. (a) J-V curves of the control and HAAc-modified PSCs. (b) J-V curves hysteresis. (c) Distribution statistics of control and HAAc-modified device efficiency. (d) The stabilized output efficiency of both the control and HAAc-treated devices. (e) EQE measurement of HAAc-modified PSCs. (f) Stability of control and HAAc-modified devices (under one-sun illumination, without encapsulation, flowing N2 atmosphere, 40℃).

This treatment acts as a chemical link connecting SnO2 and the perovskite layer, effectively neutralizing surface defects, and suppressing nonradiative recombination. The J–V curve hysteresis for both the control and HAAc-modified device is depicted in Figure 6(b), with the corresponding parameters summarized in Table S3. Following the modification, there was a notable decrease in the hysteresis index (HI) from 0.117 to 0.046. It is evident that the HAAc-modified device demonstrates minimal hysteresis. This reduction in hysteresis can primarily be ascribed to the decrease in interface defects and the inhibition of perovskite ion migration. The statistical data illustrating the distributions of efficiency for PCEs presented in Figure 6(c).

(In Supporting Information, page 7)

Table S3. The J–V curve hysteresis for the control and HAAc-modified device.

Jsc (mA/cm2)

Voc (V)

FF(%)

PCE(%)

HI

Control RS

25.23

1.103

75.27

20.9

0.117

Control FS

23.89

1.107

69.81

18.46

HAAc RS

25.49

1.190

79.63

24.16

0.046

HAAc FS

24.79

1.193

77.95

23.05

6) In Figure 6(c), it is mentioned that "The cumulative current density values for the HAAc-treated cells were calculated to be 24.11 mA cm-2, aligning well with the values derived from the J-V curves." The author should also present the External Quantum Efficiency (EQE) results and the cumulative current density values for the control group, and discuss whether these are consistent with the J-V results.

Response: We like to thank the reviewer for the valuable suggestion. We have modified the explanation of EQE results in revised version as suggested.

Action: (In manuscript, page 8,9)

Figure 6. (a) J-V curves of the control and HAAc-modified PSCs. (b) J-V curves hysteresis. (c) Distribution statistics of control and HAAc-modified device efficiency. (d) The stabilized output efficiency of both the control and HAAc-treated devices. (e) EQE measurement of HAAc-modified PSCs. (f) Stability of control and HAAc-modified devices (under one-sun illumination, without encapsulation, flowing N2 atmosphere, 40℃).

Action: (In manuscript, page 8,9)

In Figure 6(e), the External Quantum Efficiency (EQE) measurements are depicted. The integrated current density values for the control and HAAc-treated cells were calculated to be 24.02 and 24.11 mA cm-2, respectively, lightly lower than those attained from J-V curves. The EQE (External Quantum Efficiency) spectrum of a solar cell (usually ranging from 300 nm to 1100 nm) is integrated over the AM1.5G standard spectrum (according to IEC 60904-3). The EQE spectrum can be converted into the spectral response SR (?), with units of Amp/Watt, while the AM1.5G spectrum is measured in units of Watt/m2. Therefore, the integrated unit, Amp/m2, represents the current density. While the short-circuit current Jsc from IV curve of a solar cell is obtained by dividing the short-circuit current Jsc measured under STC (Standard Test Conditions) by the effective area A of the cell. Hence, the integrated Jsc from EQE is more independent and reliable. The stable-state maximum power point output, illustrated in Figure 6(d) under 1 sun for a 300-second measurement, reveals a stabilized PCE value of 20.7% for control devices (at 0.82V) and 23.8% for HAAc-treated cells (at 0.94V).

7) The author needs to clearly state the number of cells tested in Figure 6(e) and also include the standard deviation for the data presented. If only one cell each was tested for the control and HAAc-modified devices, is that representative enough to validate the results?

Response: We like to thank reviewer for pointing this out. To validate the results, we have fabricated more batches of the devices, and the related data in Figure 6(e) is summarized from at least 30 devices.

Action: (In manuscript, page 8)

The statistical data illustrating the distributions of efficiency for PCEs presented in Figure 6(c), which are summarized from at least 30 devices. It not only emphasizes the consistent performance but also highlights the significant differences between HAAc-treated cells and the control devices.